# C-Reactive Protein Is an Independent Predictor of 30-Day Bacterial Infection Post-Liver Transplantation

**DOI:** 10.3390/biom11081195

**Published:** 2021-08-12

**Authors:** Jiong Yu, Xiaowei Shi, Jing Ma, Ronggao Chen, Siyi Dong, Sen Lu, Jian Wu, Cuilin Yan, Jian Wu, Shusen Zheng, Lanjuan Li, Xiao Xu, Hongcui Cao

**Affiliations:** 1State Key Laboratory for the Diagnosis and Treatment of Infectious Diseases, The First Affiliated Hospital, Zhejiang University School of Medicine, 79 Qingchun Rd., Hangzhou 310003, China; yujiong@zju.edu.cn (J.Y.); xiaoweishi@zju.edu.cn (X.S.); 1514079@zju.edu.cn (J.M.); wj18914645776@zju.edu.cn (J.W.); yancuilin@zju.edu.cn (C.Y.); shusenzheng@zju.edu.cn (S.Z.); ljli@zju.edu.cn (L.L.); 2National Clinical Research Center for Infectious Diseases, 79 Qingchun Rd., Hangzhou 310003, China; 3Collaborative Innovation Center for the Diagnosis and Treatment of Infectious Diseases, 79 Qingchun Rd., Hangzhou 310003, China; 4NHC Key Laboratory of Combined Multi-Organ Transplantation, Key Laboratory of the Diagnosis and Treatment of Organ Transplantation, CAMS, Hangzhou 310003, China; crg_newday@163.com (R.C.); dsy20160501@163.com (S.D.); drwujian@zju.edu.cn (J.W.); 5Department of Hepatobiliary and Pancreatic Surgery, The First Affiliated Hospital, Zhejiang University School of Medicine, Hangzhou 310003, China; 6Department of Colorectal Surgery, The First Affiliated Hospital, Zhejiang University School of Medicine, Hangzhou 310003, China; 1309008@zju.edu.cn

**Keywords:** liver transplantation, systemic inflammation, C-reactive protein, MELD score, bacterial infection

## Abstract

The relationship between aseptic systemic inflammation and postoperative bacterial infection is unclear. We investigated the correlation of systemic inflammation biomarkers with 30-day clinically significant bacterial infections (CSI) after liver transplantation (LT). This retrospective study enrolled 940 patients who received LT and were followed for 30 days. The primary end point was 30-day CSI events. The cohort was divided into exploratory (*n* = 508) and validation (*n* = 432) sets according to different centers. Area under the receiver operated characteristic (AUROC) and Cox regression models were fitted to study the association between baseline systemic inflammation levels and CSI after LT. A total of 255 bacterial infectious events in 209 recipients occurred. Among systemic inflammation parameters, baseline C-reactive protein (CRP) was independently associated with 30-day CSI in the exploratory group. The combination of CRP and organ failure number showed a good discrimination for 30-day CSI (AUROC = 0.80, 95% CI, 0.76–0.84) and the results were confirmed in an external verification group. Additionally, CRP levels were correlated with bacterial product lipopolysaccharide. In conclusion, our study suggests that pre-transplantation CRP is independent of other prognostic factors for 30-day CSI post-LT, and can be integrated into tools for assessing the risk of bacterial infection post-LT or as a component of prognostic models.

## 1. Introduction

Infectious complications are the major cause of early morbidity and mortality after liver transplantation (LT) [1]. The majority of infections that occur in the month after LT are nosocomial, while most of those that develop more than one month after LT are latent and community acquired [1,2,3]. Bacterial infection is the most frequent infection complication in LT recipients, and most are caused by endogenous bacteria colonizing the oropharynx, stomach, and bowel [4]. Several risk factors are associated with infection, such as a high Model for End-stage Liver Disease (MELD) score, prolonged surgery, bleeding, choledocho-jejunostomy, dialysis, and hemorrhage [2,4].

It was recently shown that pre-LT infection is a risk factor for post-LT infection [5,6]. Patients with end-stage liver disease have high susceptibility to infection while awaiting LT, but aseptic systemic inflammation is also common in these patients [7,8,9]; it appears increased inflammatory cytokines, C-reactive protein (CRP), white blood cell count, and the higher inflammation are related to the worse outcome. Until now, whether systemic inflammation is related to postoperative bacterial infection is still unclear. The levels of systemic inflammation may be predictive of the risk of mortality in patients with cirrhosis [10,11], and are related to the long-term outcome of patients with hepatocellular carcinoma not amenable to surgery [12].

To our knowledge, no previous study has investigated the relationship between preoperative biomarkers of systemic inflammation levels and incident bacterial infection in patients after LT. In this study, we investigated the correlation of baseline biomarkers of systemic inflammation levels with 30-day clinically significant bacterial infections (CSI) post-LT.

## 2. Materials and Methods

### 2.1. Patients

This was a retrospective observational cohort study of 1261 consecutive adult patients who underwent LT between January 2015 and May 2019 at the First Affiliated Hospital, College of Medicine, Zhejiang University and Shulan (Hangzhou) Hospital. No donor organ was obtained from executed prisoners or other institutionalized persons, and all living donations were voluntary. The study was approved by the Ethics Committee of the First Affiliated Hospital, College of Medicine, Zhejiang University, and complied with the ethical guidelines of the Declaration of Helsinki. The informed consent was waived because the researchers only analyzed anonymous data.

### 2.2. Management of Patients

The LT patients were managed according to a standard protocol during follow-up. The patients received immunosuppressive therapy with basiliximab, tacrolimus, mycophenolate mofetil, and prednisolone (prednisolone was used only in patients with non-malignant end-stage liver disease or a history of rejection) [13]. Patients with ABO-incompatible LT were treated with a blood-type barrier consisting of rituximab and intravenous immunoglobulin [14]. All the patients received prophylactic antibiotics and fluconazole intravenously for 3 to 5 days. Hepatitis B immunoglobulin and a nucleos(t)ide analog were used for hepatitis B virus (HBV) prophylaxis in HBV-positive recipients. All recipients were intensively monitored for infection, rejection, and other complications according to standard protocols. Standard liver biochemical and other serological indicators were monitored regularly.

### 2.3. Preparation of Data

The following variables were collected from the electronic medical records: demographics, transplant indication, organ failure and other transplant factors, and postoperative complications. The laboratory findings were collected on the day of transplant.

### 2.4. Definition

Without knowledge of other variables, the electronic medical records, transplant database, and microbiological records were evaluated to identify clinical and laboratory-confirmed bacterial infection episodes within 30 days after transplantation. The identified infections were considered clinically significant bacterial infections (CSI) according to the standard criteria of the Centers for Disease Control and Prevention [15]. Infections were categorized as pneumonia, intra-abdominal infection, urinary tract infection, skin and soft-tissue infection, or primary bloodstream infection. The diagnostic criteria for organ failure was according to the chronic liver failure-sequential organ failure assessment (CLIF-SOFA) score [16].

### 2.5. Statistical Analysis

One center (the First Affiliated Hospital, College of Medicine, Zhejiang University) with 508 patients served as the exploratory group, and another center (Shulan (Hangzhou) Hospital) with 432 patients were used as a validation group to validate findings. The systemic inflammation biomarkers included CRP, white blood cell (WBC), neutrophil-lymphocyte ratio (NLR), and systemic immune-inflammation index (SII), and NLR = neutrophils/lymphocyte counts; SII = platelets × neutrophils/lymphocyte counts.

Continuous variables were expressed as median and interquartile range (IQR) and were compared by t-test or Mann–Whitney U test. Categorical variables were expressed as percentages and were compared by chi-squared test or Fisher’s exact test. The baseline characteristics of the patients are shown according to patients with or without CSI. Univariate and multivariate Cox proportional hazard models were performed to identify indicators of CSI patients and variables with *p <* 0.1 in a univariate analysis were then included in a forward stepwise regression model. Considering the patient’s early death without infection, a competing-risk analysis was performed to compare the association of baseline level of systemic inflammation and CSI [17]. Given that patients with infection in 30 days before LT, the presence of liver tumor, and organ failure number affects the parameters of systemic inflammation, these three subgroups were analyzed separately. Correlations between systemic inflammation and the concentration of the bacterial product lipopolysaccharide (LPS) were tested by the Spearman rank correlation test. Area under the receiver operating characteristic (AUROC) analysis was performed to assess the predictive performance of systemic inflammation biomarkers. The optimal cutoff values for biomarkers and MELD score as a predictor of mortality were determined using the Youden index. The Kaplan–Meier method was used to evaluate the cumulative rates of CSI and tested by log rank test.

Statistical analysis was performed using Statistical Package for the Social Sciences version 19.0 (International Business Machines Corporation, Armonk, NY) and R version 3.4 (R Foundation, Vienna, Austria). A two-sided *p* < 0.05 was considered to indicate statistical significance.

## 3. Results

### 3.1. Patients’ Characteristics

In total, 1261 patients were underwent LT between January 2015 and May 2019. 321 patients were excluded from the analysis, including 24 with re-transplant, seven with intra-operative arrest, nine with uncured infection before transplant, and 281 without data of CRP. Ultimately, 940 patients were included in the final analysis (Figure 1). Notably, CRP deletions in these patients occur randomly; therefore, the characteristics comparison between patients included in the analysis and patients was excluded because of the missing CRP levels, which are shown in Appendix A Appendix A.

The baseline characteristics of the patients stratified by CSI are shown in Table 1. The characteristics of the exploratory and validation group were similar, only median age is significantly higher than the validation group (50 vs. 49 years, *p* = 0.008). During the 30 days follow-up, a total of 255 CSIs occurred in 209 recipients within 30 days after LT. Of the patients, 170 had one bacterial infection, and 39 had more than one bacterial infection. As expected, significant higher levels of systemic inflammation biomarkers, MELD scores and organ failure numbers were observed in patients with 30-day CSI both in exploratory and validation groups, furthermore, the incidence of CSI in patients with liver tumors was relatively low.

### 3.2. Associations of Systemic Inflammation Levels with 30-Day CSI

Systemic inflammation biomarkers included CRP, WBC, NLR, and SII were significantly associated with 30-day CSI in the exploratory cohort, CRP (HR 1.02 [95% CI 1.02–1.03], *p* < 0.001), WBC (HR 1.06 [95% CI 1.04–1.07], *p* < 0.001), NLR (HR 1.08 [95% CI 1.06–1.11], *p* < 0.001), and SII (HR 1.0004 [95% CI 1.0002–1.0006], *p* < 0.001) (Table 2).

In the validation cohort, CRP (HR 1.02 [95% CI 1.01–1.03], *p* < 0.001), WBC (HR 1.08 [95% CI 1.05–1.11], *p* < 0.001), NLR (HR 1.03 [95% CI 1.01–1.05], *p* = 0.002), and SII (HR 1.0004 [95% CI 1.0002–1.0007], *p* < 0.001) were also associated with 30 days CSI.

### 3.3. Effect of MELD Score, Pre-LT Infection, Liver Tumor, and Organ Failure on the Level of Systemic Inflammation

As anticipated, patients with infection in 30 days and organ failure pre-transplant displayed significantly higher systemic inflammation biomarkers, but the level of CRP was not significantly different in the first three quartile intervals of the meld score, and only the fourth quartile was significantly higher (Appendix A). In addition, the levels of WBC, NLR, and SII in patients with liver tumors were significantly lower than those without liver tumors, but there was no significant difference in CRP value (Appendix A). Importantly, only CRP was a similarly good predictor of 30-day CSI in all subgroups. The predictive effect of CRP on 30-day CSI was not affected by preoperative infection, liver cancer, organ failure number, and MELD score quartiles (Table 3). Furthermore, to eliminate interference by occult and latent infections, reverse causality was evaluated by excluding patients diagnosed with infection within 1 day, 2 days, and 5 days, and similar results were obtained (Appendix A).

### 3.4. Univariate and Multivariate Analysis of Prognostic Factors for 30-Day CSI

In light of the association of CRP with independent replication associated with 30-day CSI, a multivariate stepwise Cox regression was performed to assess whether it is an independent predictor. In the multivariate analysis, the level of CRP was significantly associated with 30-day CSI (HR 1.02, 95% CI 1.01–1.03, *p* < 0.001), independently of pre-transplant factors recognized to influence outcome in the exploratory group. However, three other systemic inflammation biomarkers and MELD score were not selected as independent risk factors for 30-day CSI using a multivariable Cox proportional hazards regression model with stepwise elimination (Table 4). Considering that there is a risk of competition for patients who died in 30 days without bacterial infection, a competing-risk analysis was performed, and the correlation between 30-day CSI and baseline CRP level was consistent with the results of non-competitive models (Appendix A).

### 3.5. Prediction of 30-Day CSI

The AUC for CRP to predict 30-day CSI (0.74 [0.69–0.79]) was not significantly different from that of other markers of systemic inflammation (WBC, 0.68 [0.63–0.74], *p* = 0.104; NLR, 0.68 [0.62–0.74], *p* = 0.090) and MELD score (0.68 [0.62–0.75], *p* = 0.158), but higher than SII (0.67 [0.62–0.73], *p* = 0.038; Figure 2A, Appendix A) in the exploratory group. The stepwise regression was evaluated by the Akaike information criterion (AIC) statistics and considered as the final model with the lowest AIC. CRP and organ failure number were included in the final model, the combination of CRP with organ failure number demonstrated a trend toward greater AUROCs for 30-day CSI (CRP + organ failure numbers: AUC 0.80 [0.76–0.84]; MELD + organ failure numbers: 0.70 [0.64–0.76], *p* < 0.001; Figure 2B), and the results were confirmed in an external verification group (Figure 2C).

Application of the Youden index, the optimal cut-off value for CRP level was indicated as 9.48 mg/L (Appendix A). The Kaplan–Meier analyses graphs for 30-day CSI were based on the cut-off value for CRP and organ failure number. Patients with “high” CRP harbored significantly higher incidence of 30-day CSI (no organ failure: 3.4% vs. 26.2%; one or two organ failure: 8.3% vs. 43.9%; three or more organ failure: 60.4% vs. 80.9%) in the exploratory group, and similar results were found in the validation group (Appendix A).

### 3.6. Potential Mechanistic Role of Systemic Inflammation Levels in 30-Day CSI

In order to get a better insight into the mechanistic role of systemic inflammation parameters in patients with LT, we checked the medical record system and found that 80 patients had a concentration of the bacterial product LPS at the day of LT, the characteristics of these patients were shown in Appendix A. We compared biomarkers of systemic inflammation to markers of bacterial product. Appendix A showed a strong correlation with CRP (rho = 0.39, *p* < 0.001), but no correlation with WBC (rho = 0.14, *p* = 0.201), NLR (rho = 0.13, *p* = 0.259), and SII (rho = 0.19, *p* = 0.085). Collectively, these data suggested that CRP (positively) correlates with the bacterial translocation in patients with end-stage liver disease. The median level of LSP was significant higher in patients with postoperative CSI (0.23 vs. 0.72 EU/mL, *p* = 0.025), and elevated level of PLS was significantly associated with 30-day CSI (HR, 1.30 [95% CI 1.01–1.68], *p* = 0.041; Appendix A).

## 4. Discussion

In this respective cohort study, we report a significant association between deranged parameters of systemic inflammation before LT was significantly associated with a higher risk of 30-day CSI post-transplantation. Specifically, elevated CRP level is independently associated with increased incidence of 30-day CSI, and the association was consistent across different stratified analyses according to infection, organ failure, MELD score, and liver tumor.

Broek et al. reported the peak CRP post-LT was an independent risk factor for CSI in LT recipients, but the median duration of the observation peak values was 5 days [18], and there might already be a serious infection when the peak CRP was observed. In addition, the CRP levels of all recipients would increase non-specifically in the early stages post-LT, regardless of the development of a CSI. Therefore, better methods were needed to predict the CSI after LT. Earlier studies found that patients with preoperative infection had higher rates of postoperative infection [5,6], and patients with preoperative bacterial infection often had higher baseline CRP levels, which is in agreement with our findings. However, the risk of infection in patients with previous infection was less than twofold that in patients without infection before LT (48% vs. 25% [6], 48% vs. 30.6% [5], and 85% vs. 63% [19]). In our study, the incidence of 30-day CSI in patients with a CRP level ≥ 9.48 mg/L was 36.1%, compared to 11.5% for those with a CRP level < 9.48 mg/L, the adjusted HR of CRP level ≥ 9.48 mg/L compared to a CRP level < 9.48 mg/L was 3.16 (95% CI 2.32–4.29, *p* < 0.001). Obviously, high-risk groups can be distinguished based on their baseline CRP levels.

Our findings correspond to the results of previous cohort studies in populations with acute myeloid leukemia undergoing chemotherapy or allogeneic hematopoietic cell transplantation [20,21,22]. Sato et al. reported that a relatively high serum CRP level before the first consolidation chemotherapy was predictive of infectious events after the initial chemotherapy had reduced the interference effect of tumor cells on CRP levels, the area under the ROC curve (AUC) was 0.67 [22]. In patients with allogeneic hematopoietic cell transplantation, two studies reported a significantly greater number of infections in the high-CRP quartile compared to the low-CRP group [20,21], but other studies found that baseline CRP levels were not independently significantly predictive of infectious events [23,24], possibly due to an insufficient sample size or the effect of prophylactic antibiotics. The results of this study in a large population reinforce the connection, the CRP levels pre-LT were independently associated with 30-day CSI post-transplant.

The mechanistic role of baseline CRP levels influencing post-transplant CSI is largely unclear. Increasing evidence shows that systemic inflammation is common in patients with advanced liver cirrhosis [25,26]. Patients with liver disease often have gastrointestinal malnutrition, which leads to excessive growth of bacteria [27] and altered permeability of the intestinal barrier [28], resulting in pathological translocation of bacteria, which is the main reason for aseptic systemic inflammation [29]. Recent clinical studies found a significant reduction in infections with supply of a symbiotic consisting of one lactic acid bacterium and one fiber following major abdominal surgery or LT [30]. It has been suggested that individual fibers can reduce bacterial translocation by stimulating the growth of commensal bacteria and subsequently increasing the production of short-chain fatty acids [31], and probiotics can increase intestinal motility, stabilize the intestinal barrier, and enhance the innate immune system to suppress bacterial translocation [32,33]. This indicates that translocation of bacteria is the main cause of postoperative bacterial infection. The inflammatory response continues to stimulate circulating immune cells to release the pro-inflammatory cytokine tumor necrosis factor-α, which can disrupt intestinal tight junctions [33,34] Further augmenting intestinal permeability could contribute to enhanced translocation [35], which may explain why preoperative infection increases the risk of postoperative infection.

CRP is synthesized in the acute phase in response to inflammation, and an elevated CRP level indicates the presence of systemic inflammation [11]. In present study, the level of the bacterial product LPS was significantly associated with 30-day CSI post-LT, and LPS value was also strongly correlated with the level of CRP (rho = 0.39, *p* < 0.001). Therefore, CRP levels may reflect the degree of translocation of bacteria in patients with end-stage liver disease. Under normal conditions, viable bacteria are killed by the immune system, and their components are released into the systemic circulation, leading to systemic inflammation without overt bacterial presence. During LT, using high-dose hormones and immunosuppressants to suppress immunity permits the translocation of intestinal bacteria, leading to bacterial infection and sepsis [36]. In addition, patients with organ failure have more severe systemic inflammation [37]. In our result, the infection rate was 31.7% in patients with one or two organ failure, and reached 57.1% in patients with three or more organ failure, the rate was much higher than that in patients without organ failure (11.3%). This may support the notion that systemic inflammation increases the risk of postoperative CSI.

This was a large size observational cohort study carried out in two centers, and the findings are novel in demonstrating the association between relatively high preoperative CRP levels and an increased risk of CSI complications after LT, but several limitations must be mentioned. First, it is difficult to distinguish between occult and latent infections and translocation of bacteria. There is no guarantee that these people will not develop an occult infection. Therefore, we excluded patients who were diagnosed with CSI within 1-, 2-, and 5-days post-LT (Appendix A) and got consistent results. Second, our results show a strong correlation between baseline CRP levels and postoperative infection due to the noninvasive, inexpensive, objective, available, and widely applicable characteristics of CRP, but data on other inflammatory markers, such as procalcitonin (PCT) and IL-6, were not yet available. Third, since the LPS level changes only exist in a small proportion of patients with organ failure, a larger sample size was needed to further verify this finding.

## 5. Conclusions

In conclusion, this study reported that pre-transplantation CRP level was a novel and a potentially useful predictor of the 30-day CSI post-LT, and which positively correlated with the concentration of the bacterial product LPS in patients with end-stage liver disease. Given that CRP is already widely available in clinical routine, it could be rapidly used as a prognostic marker, which represents levels of systemic inflammation. We recommend that serum CRP can be quickly integrated into risk assessment tools before transplantation to inform clinical practice in optimize monitoring strategies.

## Figures and Tables

**Figure 1 biomolecules-11-01195-f001:**
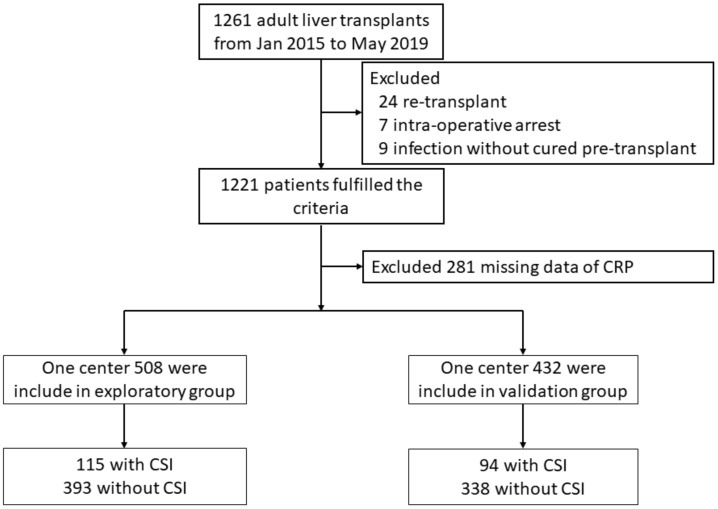
Flowchart of patient selection. CRP, C-reactive protein.

**Figure 2 biomolecules-11-01195-f002:**
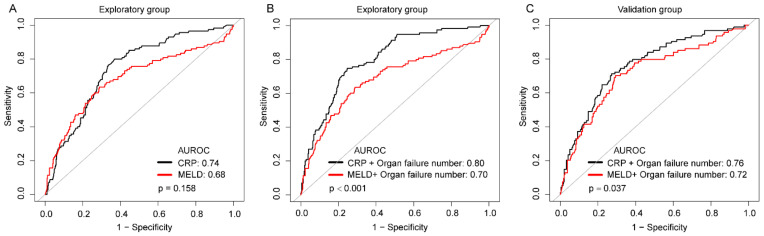
Predictive performance of CRP in 30-day CSI. (**A**) Receiver operated characteristic (ROC) curves for the established MELD score and biomarkers of systemic inflammation in relation to 30-day CSI. (**B**) Receiver operated characteristic (ROC) curves for the combination of organ failure number and CRP or MELD score in exploratory group. (**C**) Receiver operated characteristic (ROC) curves for the combination of organ failure number and CRP or MELD score in the validation group.

**Table 1 biomolecules-11-01195-t001:** Baseline characteristics of the exploratory and validation cohorts.

Variable	Exploratory Population		Validation Population	
	No CSI (*n* = 393)	CSI (*n* = 115)	No CSI (*n* = 338)	CSI (*n* = 94)
Age (years)	49 (42–57)	51 (44–60)	49 (40–55)	48 (40–56)
Female	67 (17.1%)	26 (22.6%)	53 (15.7%)	22 (23.4%)
BMI (kg/m^2^)	21.69 (19.93–24.13)	22.36 (19.71–24.24)	22.10(19.61–24.17)	22.77(20.05–25.33)
Diabetes mellitus	64 (16.3%)	20 (17.4%)	46 (13.6%)	22 (23.4%) *
MELD at transplantation	14.40 (9.90–22.88)	25.11 (15.48–30.87) **	12.97 (9.23–22.09)	24.82 (15.41–32.72) **
Sodium (mEq/L)	139 (136–141)	138 (135–141)	140 (137–142)	138 (133–140) **
Serum albumin (g/L)	35.20 (31.90–38.90)	33.20 (31.10–37.05)	34.65(31.90–38.80)	33.65(31.02–37.20)
White blood cell(10^9^/L)	4.40 (2.70–7.00)	7.80 (4.50–11.20) **	4.30 (2.70–7.07)	6.40 (4.32–10.65) **
CRP (mg/L)	7.40 (4.12–12.10)	13.90 (9.89–26.95) **	7.29 (4.00–12.89)	11.55 (7.84–25.47) **
NLR	3.50 (2.08–5.75)	6.95 (3.31–10.87) **	3.55 (1.94–6.23)	5.76 (2.90–10.44) **
SII	244 (123–476)	495 (216–913) **	231 (121–447)	421 (182–694) **
Liver tumor	171 (43.5%)	28 (24.4%) **	164 (48.5%)	27 (28.7%) **
Underlying disease				
Viral	338 (86.0%)	88 (76.5%) *	294 (87.0%)	80 (85.1%)
Alcoholic	86 (21.9%)	24 (20.9%)	67 (19.8%)	19 (20.2%)
Autoimmune disease	21 (5.3%)	9 (7.8%)	11 (3.3%)	6 (6.4%)
Other disease	15 (3.8%)	12 (10.43%) **	19 (4.5%)	3 (2.4%)
Previous abdominal surgery	82 (20.9%)	25 (21.7%)	68 (20.1%)	16 (17.0%)
Infection pre-transplant	51 (13.0%)	21 (18.3%)	35 (10.4%)	19 (20.2%) *
**Organ failure number**				
One or two	128 (32.6%)	53 (46.1%) **	77 (22. 8%)	42 (44.7%) **
Three or more	14 (3.6%)	28 (24.4%) **	25 (7.4%)	24 (25.5%) **
Donor age (years)	38 (28–48.55)	42 (32–51)	36(28–47)	41 (30–51)
Cold ischemia time (hours)	9.50 (7.30–11.80)	9.80 (7.60–12.20)	9.50 (7.00–12.00)	9.00 (7.40–11.57)
Duration of surgery (hours)	5.40 (4.70–6.10)	5.10 (4.55–6.35)	5.40 (4.70–6.20)	5.50 (4.62–6.50)
Blood loss (per 100 mL)	10 (8–18)	10 (6–19)	10 (8–16)	12 (8–20)
ABO incompatibility	50 (12.7%)	22 (19.1%)	43 (12.7%)	24 (25.5%) **
Choledocho-jejunostomy	1 (0.3%)	5 (4.4%) **	3 (0.9%)	2 (2.1%)
**Type of infection**				
Intra-abdominal	--	45 (39.1%)	--	45 (47.9%)
Pneumonia	--	59 (51.3%)	--	45 (47.9%)
Skin and soft-tissue	--	28 (24.3%)	--	17 (18.1%)
Urinary tract	--	2 (1.7%)	--	2 (2.1%)
Primary bloodstream	--	7 (6.1%)	--	6 (6.4%)

Abbreviations, CSI, clinically significant bacterial infection; CRP, C-reactive protein; NLR, neutrophil-lymphocyte ratio; SII, systemic immune-inflammation index; * *p* < 0.05; ** *p* < 0.01.

**Table 2 biomolecules-11-01195-t002:** Association between biomarkers of inflammation and 30-day CSI.

Variable	Exploratory Population		Validation Population	
	HR (95% CI)	*p*-Value	HR (95% CI)	*p*-Value
NLR	1.08 (1.06, 1.11)	<0.001	1.03 (1.01, 1.05)	0.002
WBC (10^9^/L)	1.06 (1.04, 1.07)	<0.001	1.08 (1.05, 1.11)	<0.001
SII	1.0004 (1.0002, 1.0006)	<0.001	1.0004 (1.0002, 1.0007)	0.001
CRP (mg/L)	1.02 (1.02, 1.03)	<0.001	1.02 (1.01, 1.03)	<0.001

Abbreviations, CSI, clinically significant bacterial infection; CRP, C-reactive protein; NLR, neutrophil-lymphocyte ratio; WBC, white blood cell SII, systemic immune-inflammation index; HR, hazard ratio; CI, Confidence interval.

**Table 3 biomolecules-11-01195-t003:** Association between CRP and 30-day CSI according to subgroups.

Subgroup	Events/ Patients	HR (95% CI)	*p*-Value
**Infection pre-transplant**			
No	169/814	1.02 (1.01, 1.03)	<0.001
Yes	40/126	1.02 (1.01, 1.03)	0.003
**Liver tumor**			
No	154/550	1.03 (1.02, 1.04)	<0.001
Yes	55/390	1.02 (1.01, 1.03)	<0.001
**Organ failure number**			
None	62/549	1.02 (1.02, 1.03)	<0.001
One or two	95/300	1.02 (1.01, 1.03)	0.001
Three or more	52/91	1.02 (1.00, 1.03)	0.014
**MELD score** **quartile**			
Quartile 1	30/222	1.02 (1.01,1.03)	0.003
Quartile 2	20/243	1.02 (1.00,1.04)	0.048
Quartile 3	59/235	1.03 (1.02, 1.05)	<0.001
Quartile 4	100/240	1.02 (1.02, 1.03)	<0.001

Abbreviations, CRP, C-reactive protein; HR, hazard ratio; CI, Confidence interval; CSI, clinically significant bacterial infection.

**Table 4 biomolecules-11-01195-t004:** Univariate and multivariate analysis of risk factors for 30-day CSI in exploratory group.

Variable	Univariate Analysis		Multivariate Analysis	
	HR (95% CI)	*p*-Value	HR (95% CI)	*p*-Value
Liver tumor	0.46 (0.30, 0.70)	<0.001		
MELD at transplantation	1.07 (1.05, 1.09)	<0.001	1.03 (0.99, 1.07)	0.096
Organ failure number (None)	Reference		Reference	
One or two	2.69 (1.75, 4.14)	<0.001	1.47 (0.74, 2.93)	0.276
Three or more	8.81 (5.32, 14.59)	<0.001	3.49 (1.39, 8.75)	0.008
Donor age (years)	1.01 (1.00, 1.03)	0.082		
ABO incompatibility	1.55 (0.98, 2.47)	0.064		
Choledocho-jejunostomy	6.27 (2.55, 15.42)	<0.001	8.04 (3.21, 20.15)	<0.001
Serum albumin (g/L)	0.97 (0.94, 1.00)	0.082		
NLR	1.08 (1.06, 1.11)	<0.001	1.03 (1.00, 1.06)	0.079
SII	1.0004 (1.0003, 1.0006)	<0.001		
WBC (10^9^/L)	1.06 (1.04, 1.07)	<0.001		
CRP (mg/L)	1.02 (1.01, 1.03)	<0.001	1.02 (1.01, 1.03)	<0.001

Abbreviations, CSI, clinically significant bacterial infection; CRP, C-reactive protein; NLR, neutrophil-lymphocyte ratio; WBC, white blood cell; SII, systemic immune-inflammation index; HR, hazard ratio; CI, Confidence interval.

## Data Availability

The data presented in this study are available on reasonable request from the corresponding author.

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
