# Peer review of "C-Reactive Protein Is an Independent Predictor of 30-Day Bacterial Infection Post-Liver Transplantation"

_biomolecules, 2021, doi:10.3390/biom11081195_

Round 1
Reviewer 1 Report
I read with interest the manuscript entitled “C-reactive protein is an independent predictor of 30-day bacterial infection post-liver transplantation ” by Yiong Yu et al. In this manuscript, the authors conducted a retrospective analysis of 940 adult liver transplants.
The aim of this study was to investigate the correlation of systemic inflammation biomarkers with 30-day clinically significant bacterial infections (CSI) after LT. This article is well written but is difficult to read especially in the results section.
The subject is interesting but I remained concerned by the impact of the study and I consider that some points should be reworked. The definition of many parameters (exploratory group, validation group, Systemic immune inflammatory index ..) is missing.
The authors explore the correlation of baseline biomarkers of systemic inflammation levels : what does “baseline” mean ? just prior LT, within one week, one month ? It might be interesting to explore the evolution of the biomarkers of systemic inflammation levels during the first days after liver transplantation. In addition, I do not understand the separation into 2 subgroups (validation and exploratory) of the cohort. The authors are not trying to validate a score, just to confirm what they found.
An analysis with an external cohort is more interesting. The selection of patients in sub-group is not clear. The authors need to simplify the flow chart. The manuscript, in particular, the statistical section and result section is confusing. In the statistical section the authors state to use an Univariate and multivariate Cox proportional hazard models to identify indicators of CSI patients. In the results section, table 5, the authors describe the results of Multivariate logistic regression analyses. Why ? I advise the authors to redo the statistical analysis with a statistician. I understand that the authors' objective is to show that inflammation is a key factor in the prognosis of LT patients. However, is the method used comparing these markers the right one ? More information about bacterial infection, prophylactic antibiotics, cause of death would be helpful to the readers to understand the impact of inflammation on the prognostic of patients included in this study. The paper is interesting for a publication. However, I invite the authors to answer the following questions in order to make the article useful in clinical practice: For example, which patient should I propose for transplantation ? should the duration and prophylactic antibiotics be modified ? To answer these questions, authors should review the design of their study and propose for example a decision tree based on systemic inflammation biomarkers levels.
Reviewer 2 Report
Yu et al conducted a retrospective study to evaluate the impact of pre-transplant CRP serum value to stratify patients who are at risk for infection within 30 days after transplantation.
Nine-hundred forty patients were recruited for the study, where n=508 served for exploration and n= 432 patients were used for validation.
Among the inflammation parameters, CRP was the best independent predictor for infection within 30 days after transplantation. This result was confirmed in the validation cohort.
There are some concerns:
- The authors found that the Youden Index was 9,79 mg/dl, why 10 mg/dl for CRP was accepted as a cut-off value.
- First question how was the sensitivity and specificity for infection?
- How was the postoperative infection rate for both cohorts, when CRP < 10 mg compared to CRP > 10 mg/dl?
- Postoperative infection is associated with several issues, like
- early allograft dysfunction (EAD)
- Blood loss; in one table the blood loss is given with 100 ml, which occurs in only 10 patients?? Is that really correct?
- Postoperative dialysis-dependent kidney failure, how was the dialysis rate postop.?
- Discussion: Concern CRP and infection, van den Broek et al. (Liver Transplantation 2010, 402-410) evaluated CRP in the post-transplant course as a marker for infection. The values were close to yours, in patients with infection CRP was > 10 mg/dl, and those without had a CRP < 10 mg/d. Please add this paper to your discussion and comment on it.
Round 2
Reviewer 1 Report
I have read again this retrospective multicenter cohort study by Jiong Yu and colleagues
The authors replied to almost all comments and the manuscript is clearly improved. The results are simplified and easier to understand especially Table 4.
Page 3 Line 97, Please add the name of center (exploratory group and validation group)
Please be sure to spell out all abbreviations when first used.
Figure 2A, please remove the ROC curves for WBC, NLR and SII. the aim being to have more clarity in the figure. These results are available in the table S8.
Figure 2B: A statistical comparison of the ROC curves could be interesting
The English is good, but there remains some mistakes. If possible, review by a native speaker would be helpful.
Author Response
We appreciate the reviewer’s suggestions and insightful comments.

Reviewer 2 Report
The authors addressed all the raised issues
Author Response
We appreciate the reviewer’s suggestions and insightful comments. The manuscript has been carefully reviewed by a colleague whose native language is English. We have revised the grammar, layout, etc. of the manuscript. All changes to the content are indicated in red in the revised manuscript.
